# Impact of asymptomatic *Plasmodium falciparum* infection on the risk of subsequent symptomatic malaria in a longitudinal cohort in Kenya

Kelsey M Sumner[1,2], Judith N Mangeni[3], Andrew A Obala[4], Elizabeth Freedman[2], Lucy Abel[5], Steven R Meshnick[1‡], Jessie K Edwards[1], Brian W Pence[1], Wendy Prudhomme-O'Meara[2,3,6†], Steve M Taylor[1,2,6†]*

[1]Department of Epidemiology, Gillings School of Global Public Health, University of North Carolina, Chapel Hill, United States; [2]Division of Infectious Diseases, School of Medicine, Duke University, Durham, United States; [3]School of Public Health, College of Health Sciences, Moi University, Eldoret, Kenya; [4]School of Medicine, College of Health Sciences, Moi University, Eldoret, Kenya; [5]Academic Model Providing Access to Healthcare, Moi Teaching and Referral Hospital, Eldoret, Kenya; [6]Duke Global Health Institute, Duke University, Durham, United States

*For correspondence:
steve.taylor@duke.edu

†These authors contributed equally to this work

‡Deceased

**Competing interests:** The authors declare that no competing interests exist.

## Abstract

**Background:** Asymptomatic *Plasmodium falciparum* infections are common in sub-Saharan Africa, but their effect on subsequent symptomaticity is incompletely understood.
**Methods:** In a 29-month cohort of 268 people in Western Kenya, we investigated the association between asymptomatic *P. falciparum* and subsequent symptomatic malaria with frailty Cox models.
**Results:** Compared to being uninfected, asymptomatic infections were associated with an increased 1 month likelihood of symptomatic malaria (adjusted hazard ratio [aHR]: 2.61, 95% CI: 2.05 to 3.33), and this association was modified by sex, with females (aHR: 3.71, 95% CI: 2.62 to 5.24) at higher risk for symptomaticity than males (aHR: 1.76, 95% CI: 1.24 to 2.50). This increased symptomatic malaria risk was observed for asymptomatic infections of all densities and in people of all ages. Long-term risk was attenuated but still present in children under age 5 (29-month aHR: 1.38, 95% CI: 1.05 to 1.81).
**Conclusions:** In this high-transmission setting, asymptomatic *P. falciparum* can be quickly followed by symptoms and may be targeted to reduce the incidence of symptomatic illness.
**Funding:** This work was supported by the National Institute of Allergy and Infectious Diseases (R21AI126024 to WPO, R01AI146849 to WPO and SMT).

## Introduction

Asymptomatic *Plasmodium falciparum* infections, defined as the presence of parasites in the absence of symptoms, are common across sub-Saharan Africa. In 2015, a geo-spatial meta-analysis estimated a continent-wide prevalence of asymptomatic *P. falciparum* in children aged 2 to 10 years of 24% based on microscopy and rapid diagnostic test (RDT) results (*Snow et al., 2017*). In high-transmission settings, these infections are more common, with point prevalence among adults exceeding 30% in the Democratic Republic of the Congo (*Taylor et al., 2011*) and Malawi (*Topazian et al., 2020*). Though by definition lacking acute symptomatology, these infections when persistent can adversely affect the individual (*Cottrell et al., 2015*; *Maketa et al., 2015*;

*Matangila et al., 2014*; *Sifft et al., 2016*) as well as serve as a reservoir for onward parasite transmission (*Gouagna et al., 2004*; *Tadesse et al., 2018*).

Although epidemiologically important, the natural history of asymptomatic *P. falciparum* and its relationship to future symptomatic malaria remains unclear. In prior studies, asymptomatic *P. falciparum* infections have been observed to both decrease (*Buchwald et al., 2019*; *Males et al., 2008*; *Portugal et al., 2017*; *Sondén et al., 2015*) and increase (*Le Port et al., 2008*; *Liljander et al., 2011*; *Njama-Meya et al., 2004*; *Nsobya et al., 2004*) the risk of symptomatic malaria, and several have correlated this heterogeneity to people's age or site transmission intensity (*Henning et al., 2004*; *Wamae et al., 2019*). Inferences have been further complicated in these studies owing to their cross-sectional capture of asymptomatic infections (*Henning et al., 2004*; *Liljander et al., 2011*; *Males et al., 2008*; *Nsobya et al., 2004*; *Sondén et al., 2015*; *Wamae et al., 2019*), short follow-up periods (*Le Port et al., 2008*; *Njama-Meya et al., 2004*), or limited age ranges (*Le Port et al., 2008*; *Liljander et al., 2011*; *Males et al., 2008*; *Njama-Meya et al., 2004*; *Nsobya et al., 2004*; *Wamae et al., 2019*), which collectively undermine a clear understanding of the risk of symptomatic malaria following the detection of an asymptomatic infection.

We investigated the natural history of asymptomatic *P. falciparum* infections in a high-transmission setting using a 29-month longitudinal cohort of people aged 1 to 85 years in Western Kenya. Using monthly active case detection of asymptomatic infections and passive capture of symptomatic events, we evaluated the likelihood of symptomatic malaria following an asymptomatic *P. falciparum* infection. We hypothesized that infection with asymptomatic parasitemia would be associated with a decrease in future risk of symptomatic malaria compared to uninfected people and that, because age serves as a proxy for prior cumulative exposure, this effect would be most pronounced in older people.

## Materials and methods

### Study population, sample collection, and sample processing

From June 2017 to November 2019, we followed a cohort of 268 people aged 1 to 85 years living in 38 households in a rural setting in Webuye, Western Kenya (*O'Meara et al., 2020*). The cohort was assembled using radial sampling of 12 households per village for three villages with high malaria transmission. The first household in each village was randomly selected. Two households moved during follow-up and were replaced. For each person, asymptomatic *P. falciparum* infections were detected monthly by active surveillance through collecting questionnaires and dried blood spot (DBS) samples for post hoc molecular parasite detection. Symptomatic *P. falciparum* infections were detected using passive surveillance by testing people with self-reported symptoms with a malaria RDT (Carestart Malaria HRP2 *Pf* from Accessbio) and collecting a DBS (*AccessBio, 2019*). People with positive RDT results were treated with Artemether-Lumefantrine (AL).

DBS were processed to detect *P. falciparum* infections by extracting genomic DNA (gDNA) from DBS and then tested in duplicate for *P. falciparum* parasites using a duplex real-time PCR (qPCR) assay targeting the *P. falciparum pfr364* motif and human β-tubulin gene (*Plowe et al., 1995*; *Taylor et al., 2019*). From each DBS, three individual punches were deposited in a single well of a 96-well deep well plate and extracted with Chelex-100 following Saponin and Proteinase K treatments. gDNA was ultimately preserved in approximately 100 µL of solution. Each gDNA extract was tested in duplicate. Each reaction contained 2 µL of gDNA template in a 12 µL total reaction volume, and templates were tested in 384-well plates on an ABI QuantStudio6 platform. Samples were defined as *P. falciparum*-positive if: (i) both replicates amplified *P. falciparum* and both Ct values were < 40 or (ii) one replicate amplified *P. falciparum* and the Ct value was < 38.

Parasite densities were estimated using standard curves generated from amplifications on each plate from templates of known parasite density. To generate these templates, parasite strain 3D7 was cultivated in vitro using standard conditions and the parasite density was estimated initially by light microscopy of Giemsa-stained slides and then by hemocytometer. For the latter, after averaging estimates of parasite density from five to six chambers, the non-diluted sample was diluted with fresh whole blood to obtain a 2000 p/µL stock solution. This was then serially diluted with whole blood to obtain 1000, 200, 100, 20, 10, 2, 1, 0.2, and 0.1 p/µL samples. Each of these was then prepared as a DBS of 30 µL volume, and from these gDNA was extracted as above for clinical samples.

The result was a series of gDNA samples from mocked templates of known concentrations of 3D7 parasite that were processed identically to the clinical samples.

## Exposure and outcome ascertainment

The main exposure was an asymptomatic *P. falciparum* infection during monthly active case detection assessments, defined as *P. falciparum*-positive by qPCR in a person lacking symptoms. People who were *P. falciparum*-negative by qPCR during monthly visits were considered uninfected. Participant follow-up was imputed for the first consecutive missed monthly visit during each follow-up period by carrying forward the previous month's value as the exposure status of the missed monthly visit (*Nguyen et al., 2018*). If a person missed two or more consecutive monthly visits, they were considered lost to follow-up and censored at the time of the imputed monthly visit. A sensitivity analysis was conducted for imputation using a dataset without imputation for missed monthly visits. Participants were allowed to enter and leave the study throughout the study period. At the end of the study period, all participants were censored.

The main outcome assessed was days to symptomatic malaria. We defined symptomatic *P. falciparum* infection as the current presence of at least one symptom consistent with malaria during a sick visit (i.e. fever, aches, vomiting, diarrhea, chills, cough, or congestion) and *P. falciparum*-positive by both RDT and qPCR. Outcome events occurring within 14 days of receipt of AL for a symptomatic infection were excluded.

Some participants were classified as symptomatically infected at a monthly visit through passive detection of symptoms; this occurred if a study team member conducting a monthly visit was approached by a participant reporting malaria-like symptoms. The study team member would then perform an RDT and record information as for a passively detected sick visit. Symptoms were not routinely elicited during interviews on monthly visits. If the person met the case definition for symptomatic malaria, they were confirmed as having a symptomatic visit on that day and removed from follow-up until 14 days post-receipt of antimalarials or the next monthly follow-up visit. If that person did not meet our case definition for symptomatic malaria, then they were removed from follow-up for that month and re-entered for follow-up in the following month.

## Hazard of symptomatic malaria analysis

Across all participants, we estimated the hazard of subsequent symptomatic malaria when infected with asymptomatic malaria compared to being uninfected at monthly visits. The hazard of symptomatic malaria was calculated for multiple follow-up periods: (i) 1 month, (ii) 3 months, (iii) 6 months, (iv) 12 months, and (v) 29 months (entire study period). For each follow-up period exceeding 1 month, exposure status was ascertained at every monthly follow-up visit and allowed to vary each month using a method proposed by *Hernán et al., 2005*. This method treats each monthly follow-up visit as a new study entry, recalculating the time to symptomatic malaria or censoring using each monthly follow-up visit date as the origin and attributing the exposure in that month as the exposure status from that month up until the event or censoring (*Figure 1*). This exposure coding method was chosen due to its ability to capture the exposure at multiple time points with less risk of misclassification or left truncation bias compared to alternative time-varying coding approaches (*Supplementary file 1*).

## Statistical modeling

We first estimated the time to symptomatic malaria across the full 29 months using Kaplan-Meier curves and the log-rank test. We compared differences in median time to symptomatic malaria across select covariates using the Wilcoxon rank sum test with continuity correction for dichotomous variables or the Kruskal-Wallis test for polytomous variables. The Bonferroni correction was applied to all table p-values to account for repeated measures during the 29 months of follow-up.

In order to account for anticipated confounders of the relationship between asymptomatic infection and symptomatic malaria, we next computed a multivariate frailty Cox proportional hazards model (*Equation 1*).

$$\frac{h_1(t)_i}{h_0(t)_i} = \exp(\alpha_i + \beta_1 Asymptomatic\ infection_{im} + \beta_2 Age\ 5\ to\ 15_i + \beta_3 Age\ over\ 15_i + \beta_4 Female_i + \beta_5 Regular\ bed\ net\ usage_i + \beta_6 village:Maruti + \beta_7 village:sitabicha + \epsilon_i)$$

(1)

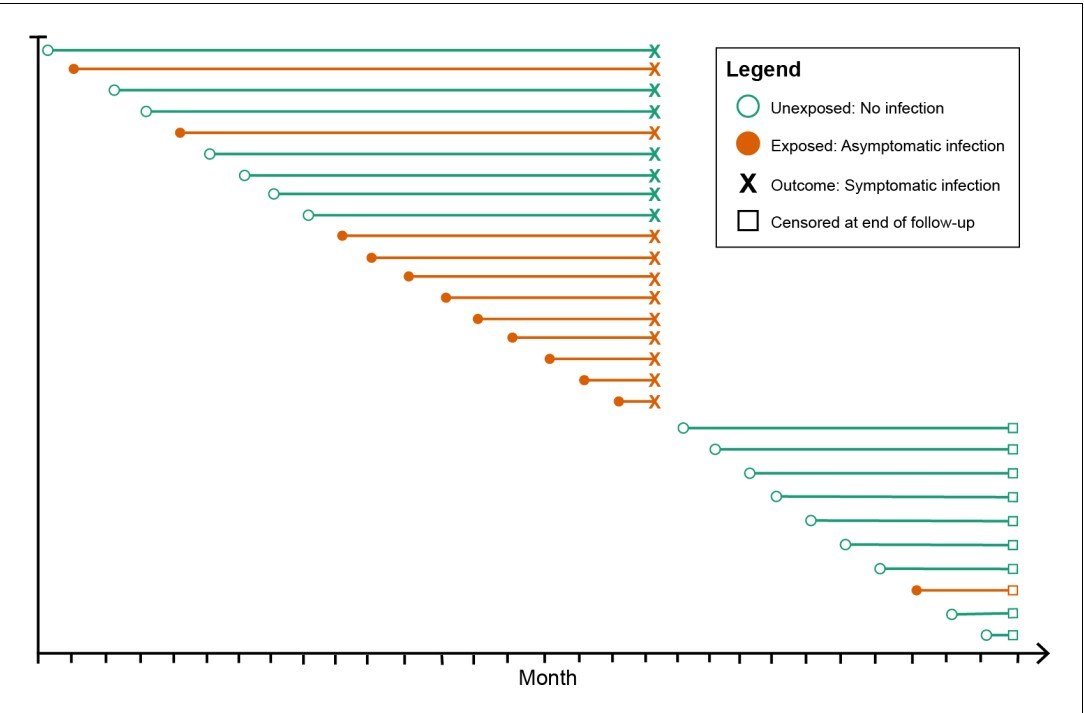

**Figure 1.** Schematic of how asymptomatic exposure status was ascertained for one participant's follow-up. The method treated each monthly follow-up visit as a new study entry for the participant (denoted by circles), recalculating the time to symptomatic malaria (denoted by X) using the monthly follow-up visit date as the origin. The exposure status for each monthly follow-up visit became the exposure status for the follow-up period (denoted by the horizontal line). The follow-up period ended if the participant had a symptomatic infection (X) or was censored due to the study ending or becoming lost to follow-up (denoted by squares). Illustrated here is one hypothetical participant's follow-up during 29 months, during which they contributed follow-up periods after being uninfected (green lines) and after having asymptomatic infections (orange lines), and suffered one episode of symptomatic malaria (denoted by X), before being censored at the close of follow-up (denoted by squares). As a result, each participant contributes multiple entries to each model equal to the number of exposure assessments, and models include a random effect at the level of the participant to account for repeated observations.

The model controlled for the following confounders as determined by a directed acyclic graph (*Figure 3—figure supplement 1*): age (<5 years, 5 to 15 years, >15 years), sex, and regular bed net usage (averages > 5 nights a week sleeping under a bed net – yes, no). To account for differences in malaria prevalence across the three villages, we also included a covariate in the model to represent each village. We allowed the main exposure to vary each month based on the monthly follow-up visit infection status ($m$), and included a random intercept at the participant level ($\alpha_i$) to account for potential correlated intra-individual outcomes. A log-normal distribution was used for the random effect. $\epsilon_i$ represented the model's error term. Additional models incorporated either an additional random effect at the household level or a robust error estimator at the participant level. The proportional hazards assumption was assessed using Kaplan-Meier curves and Schoenfeld residual plots.

We tested for effect measure modification by age and sex by stratifying the multivariate model by age category (<5 years, 5 to 15 years, >15 years) or sex, computing hazard ratios and 95% confidence intervals (CI) of the main exposure, and comparing a Cox proportional hazards model with an interaction term between the potential modifier and main exposure to *Equation 1* using the log-likelihood ratio test.

We computed an additional time-to-event model using a subset of events. Because asymptomatic infections could represent incipiently symptomatic (i.e. 'pre-symptomatic') infections, we excluded all monthly follow-up visits occurring within 14 days prior to a symptomatic infection, reducing the possibility that pre-symptomatic infections could be misclassified as asymptomatic. The time frame for identifying potentially pre-symptomatic infections was chosen for consistency with previous work

studying time to symptomatic malaria (*Buchwald et al., 2019*). The analysis was conducted using *Equation 1* for the 1-, 3-, 6-, 12-, and 29-month follow-up periods. All statistical analyses were performed using R version 4.0.2 (*R Development Core Team, 2020*) with the packages tidyverse (*Wickham et al., 2019*), survminer (*Kassambara et al., 2020*), survival (*Therneau and Grambsch, 2000*), coxme (*Therneau, 2020*), lme4 (*Bates et al., 2015*), and ggalluvial (*Brunson, 2020*). Code is available on Github: https://github.com/duke-malaria-collaboratory/time_to_symptomatic_malaria, (*Sumner, 2021a*; copy archived at swh:1:rev: 95b7f8268baa6007af84cc7ee0f110f2a3629631, *Sumner, 2021b*). Statistical significance was assessed at an $\alpha$ level of 0.05.

## Detectability of asymptomatic infections

Asymptomatic infections defined as above were further classified as meeting a series of thresholds of parasite densities: any density, >1, >10, >100, >500, and >1000 parasites/µL. These classifications were assigned in a non-mutually exclusive fashion to asymptomatic infections, and then the 1-month likelihoods of symptomatic malaria relative to uninfected people were modeled separately using the Cox proportional hazards model in *Equation 1*. As an additional analysis, we repeated this process for each parasite density threshold stratified by participant age (<5 years, 5 to 15 years, >15 years).

## Sensitivity analyses

We computed sensitivity analyses to account for potential misclassification of the outcome of symptomatic malaria in the main models over 1- and 29-month intervals. A 'permissive' case definition defined a symptomatic infection as one where a participant had at least one symptom consistent with malaria during a sick visit and was *P. falciparum*-positive by real-time PCR (qPCR). A 'stringent' case definition defined a symptomatic infection as one where a participant had a self-reported fever during a sick visit and was *P. falciparum*-positive by both RDT and qPCR. Additional sensitivity analyses were computed to investigate the separate effects of additional covariates by incorporating into the frailty Cox proportional hazards model in *Equation 1*, a new term for the covariate of interest. For seasonality, we classified monthly visits that occurred any time from May to October as the high-transmission season and from November to April as the low-transmission season, based on the region's rainy seasons. For the number of prior infections, we included in the model as a covariate the number of prior infections as a continuous number. For prior antimalarial treatment, we included a variable coded dichotomously as having received study-prescribed antimalarials up until that monthly visit or not; a person was coded as having not received study-prescribed antimalarials up until their first symptomatic infection, but afterward were coded as receiving treatment from that point forward in follow-up.

## Ethical review

The study was approved by institutional review boards of Moi University (2017/36), Duke University (Pro00082000), and the University of North Carolina at Chapel Hill (19-1273). All participants or guardians provided written informed consent, and those over age 8 years provided additional assent.

## Results

For 29 months, we followed 268 participants from three villages in Western Kenya. After excluding participants with less than 2 months of follow-up, the analysis dataset consisted of 257 participants with a median of 222 days (interquartile range [IQR]: 89, 427) of follow-up and a median age of 13 years (range: 1, 85) (*Figure 3—figure supplement 2*). Overall, 5379 person-months at risk were observed with 1842 (34.2%) person-months of asymptomatic malaria exposure; the median total months of asymptomatic exposure for a participant was 9 (IQR: 5, 17). Exposure status frequently changed for participants and remained constant for only 16 (6.2%) people across follow-up; four people were asymptomatically infected for the entirety of follow-up and only 12 people were never infected (*Figure 2A*). We recorded 266 symptomatic malaria events. Participants had a median of 1 (IQR: 0, 2) symptomatic infection during follow-up. Median time to symptomatic malaria when asymptomatically infected (173, IQR: 49, 399) was shorter than when uninfected (230, IQR: 98, 402), as well as shorter for participants aged 5 to 15 years or living in the village Maruti (*Table 1*).

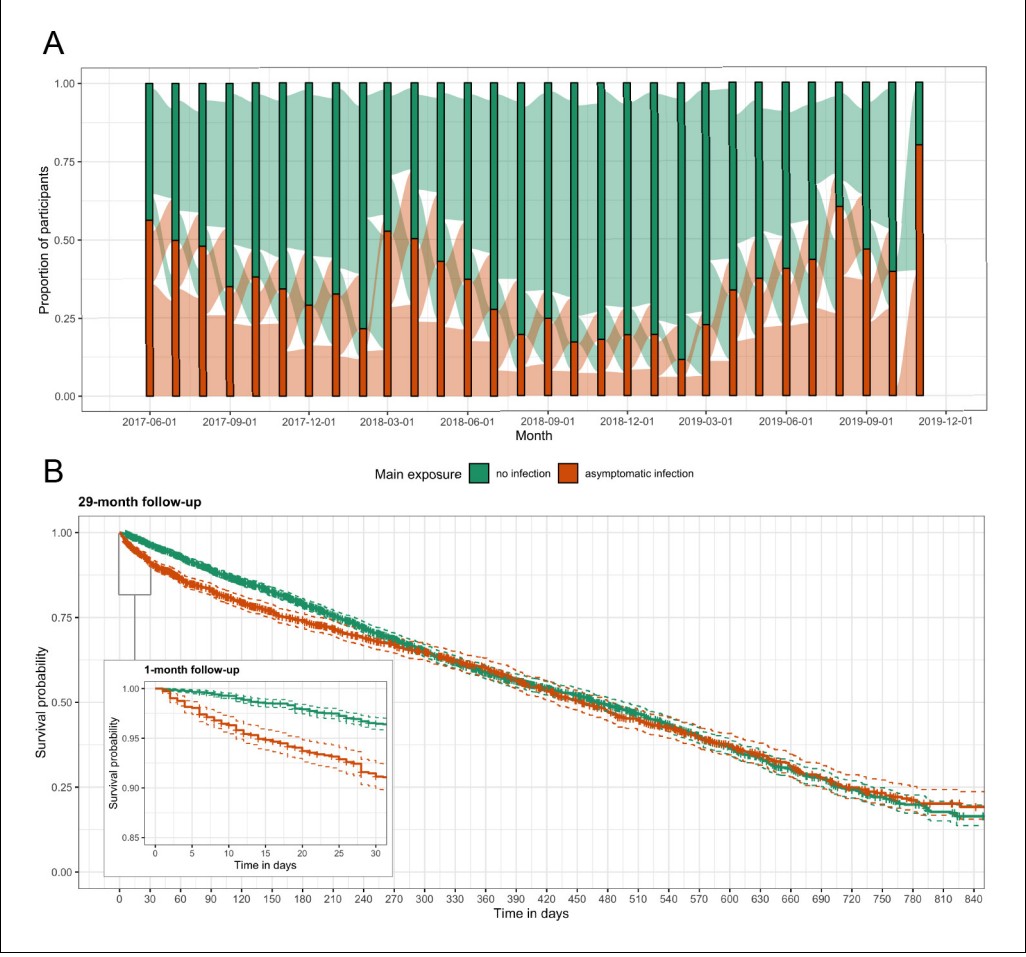

**Figure 2.** Asymptomatic malaria exposure classifications and symptomatic malaria outcomes over time. (**A**) The proportion of participants who had either an asymptomatic infection (orange) or were uninfected (green) at each monthly visit is indicated by the bars. The ribbons connecting the bars illustrate the proportion of participants who moved exposure status from month to month. Orange ribbons indicate the proportion of participants with asymptomatic infections and green the proportion that were uninfected. (**B**) A Kaplan-Meier survival curve of symptomatic events across the follow-up period stratified by asymptomatic malaria exposure is displayed. Each exposure-outcome pair depicted in *Figure 1* is plotted, and therefore each study participant is registered in the curve multiple times, equal to the number of exposure classifications.

Comparison of Kaplan-Meier curves over the full 29 months indicated a difference in the time to symptomatic malaria in the first few months post asymptomatic infection but not long term (p-value = 0.100 by log-rank test) (*Figure 2B*). Results for secondary case definitions for symptomatic malaria were overall similar and are provided in the supplement (*Supplementary files 2* and *3*).

## Short-term effect of asymptomatic malaria exposure

In a univariate frailty Cox proportional hazards model, compared to uninfected people, the 1-month crude hazard ratio of symptomatic malaria for participants with asymptomatic infections was 2.69 (95% CI: 2.12 to 3.43). This association was similar in a model adjusted for covariates (adjusted HR [aHR]: 2.61, 95% CI: 2.05 to 3.33) (*Table 2*, *Figure 3A*) as well as when using alternative modeling approaches, alternate outcome case definitions, and in sensitivity analyses. This relationship between asymptomatic malaria and subsequent symptomatic illness was not modified by age (p-value = 0.447 by log-likelihood ratio test), because asymptomatic infections were associated with significantly increased likelihoods of subsequent symptomatic malaria in all age categories: <5 years (aHR: 3.77, 95% CI: 2.02 to 7.04), 5 to 15 years (aHR: 2.45, 95% CI: 1.79 to 3.35), and >15 years (aHR: 2.55, 95% CI: 1.57 to 4.15) (*Table 3*). In contrast, sex did modify this relationship (p-value = 0.006 by

**Table 1.** Covariate distribution across symptomatic malaria events in 29 months of follow-up.

| | Total person-months[†] (N, %) | Person-months ending in symptomatic infections[‡] (N, %) | Median time to symptoms for entire study (days, IQR) | p-Value comparing time to symptoms |
|---|---|---|---|---|
| Main exposure | | | | **<0.001**[§] |
| No infection | 3537 (65.8) | 1580 (65.7) | 230 (98, 402) | – |
| Asymptomatic infection | 1842 (34.2) | 826 (34.3) | 173 (49, 399) | – |
| Age | | | | **0.015**[¶] |
| <5 years | 812 (15.1) | 329 (13.7) | 226 (82, 435) | – |
| 5 to 15 years | 2279 (42.4) | 1319 (54.8) | 199 (70, 379) | – |
| >15 years | 2288 (42.5) | 758 (31.5) | 244 (97, 426) | – |
| Sex | | | | 0.779[§] |
| Male | 2360 (43.9) | 1190 (49.5) | 229 (86, 420) | – |
| Female | 3019 (56.1) | 1216 (50.5) | 202 (76, 384) | – |
| Regular bed net usage[*] | | | | 1.000[§] |
| No | 1425 (26.5) | 730 (30.3) | 210 (82, 386) | – |
| Yes | 3954 (73.5) | 1676 (69.7) | 217 (80, 403) | – |
| Village | | | | **<0.001**[¶] |
| Kinesamo | 1854 (34.5) | 876 (36.4) | 233 (89, 418) | – |
| Maruti | 1681 (31.3) | 745 (31.0) | 174 (64, 350) | – |
| Sitabicha | 1844 (34.3) | 785 (32.6) | 231 (90, 421) | – |

Abbreviations: IQR, interquartile range.

[*]Regular bed net usage was a person averaging > 5 nights a week sleeping under a bed net.

[†]Total person-months indicates the total number of monthly follow-up visits ending in a symptomatic infection or censoring for full 29 months of follow-up.

[‡]Symptomatic infections were defined using the primary case definition where a participant was *P. falciparum*-positive by both RDT and qPCR as well as had at least one symptom consistent with malaria during a sick visit.

[§]Wilcoxon rank sum test with continuity correction and Bonferroni correction for repeated measures.

[¶] Kruskal-Wallis test with Bonferroni correction for repeated measures.

Significant estimates are bolded.

log-likelihood ratio test) (*Table 3*), whereby the risk of symptomatic malaria following asymptomatic infection was lower for males (aHR: 1.76, 95% CI: 1.24 to 2.50) compared to females (aHR: 3.71, 95% CI: 2.62 to 5.24) (*Figure 3—figure supplement 3*). We observed similar 1-month elevated risks of malaria in asymptomatically infected people when using both the 'permissive' (aHR 1.97, 95% CI: 1.63 to 2.40) and the 'stringent' (aHR 2.76, 95% CI: 2.11 to 3.62) alternate case definitions for symptomatic malaria (*Figure 3—figure supplement 3*).

In a subset analysis accounting for potentially pre-symptomatic infections, compared to uninfected people, the risk of symptomatic malaria was increased in those with asymptomatic infections by more than 1.7 times (aHR: 1.77, 95% CI: 1.26 to 2.47) when limited to those with events more than 14 days after exposure ascertainment (*Figure 3A*). In this subset, we did not observe effect measure modification by participant age or sex (*Table 3*). The 1-month adjusted risk of symptomatic malaria was not substantially different in models incorporating seasonality (aHR: 2.46, 95% CI: 1.93 to 3.15), the number of prior asymptomatic infections (aHR: 2.60, 95% CI: 2.03 to 3.31), or prior anti-malarial treatment (aHR: 2.61, 95% CI: 2.05 to 3.33), nor in a model using the dataset without imputation (aHR: 2.75, 95% CI: 2.05 to 3.66).

## Long-term effect of asymptomatic malaria exposure

Next, we assessed the relationship between asymptomatic infection and subsequent symptomatic malaria over longer follow-up periods. Extending the follow-up period led to a diminution in the risk of symptomatic malaria comparing those asymptomatically infected versus uninfected over 3 months

**Table 2.** Predicted hazard of symptomatic malaria across follow-up periods.

| | 1-Month aHR (95% CI) | 3-Month aHR (95% CI) | 6-Month aHR (95% CI) | 12-Month aHR (95% CI) | 29-Month aHR (95% CI) |
|---|---|---|---|---|---|
| **Main exposure** | | | | | |
| No infection | Ref | Ref | Ref | Ref | Ref |
| Asymptomatic infection | **2.61 (2.05, 3.33)** | **1.64 (1.40, 1.94)** | **1.38 (1.20, 1.58)** | 1.12 (1.00, 1.25) | **1.11 (1.01, 1.22)** |
| **Age** | | | | | |
| <5 years | Ref | Ref | Ref | Ref | Ref |
| 5 to 15 years | 1.37 (0.90, 2.08) | 1.61 (1.00, 2.61) | **1.99 (1.07, 3.71)** | 2.37 (0.97, 5.77) | **2.52 (1.26, 5.01)** |
| >15 years | **0.56 (0.36, 0.88)** | 0.74 (0.46, 1.21) | 0.83 (0.44, 1.53) | 0.88 (0.37, 2.08) | 0.97 (0.51, 1.84) |
| **Sex** | | | | | |
| Male | Ref | Ref | Ref | Ref | Ref |
| Female | 0.93 (0.70, 1.24) | 0.84 (0.61, 1.16) | 0.80 (0.53, 1.20) | 0.68 (0.38, 1.21) | **0.63 (0.40, 0.99)** |
| **Regular bed net usage*** | | | | | |
| No | Ref | Ref | Ref | Ref | Ref |
| Yes | 1.00 (0.70, 1.43) | 0.81 (0.55, 1.20) | 0.70 (0.43, 1.16) | 0.59 (0.29, 1.21) | **0.52 (0.30, 0.89)** |
| **Village** | | | | | |
| Kinesamo | Ref | Ref | Ref | Ref | Ref |
| Maruti | 1.08 (0.77, 1.52) | 1.11 (0.75, 1.64) | 1.14 (0.69, 1.88) | 1.13 (0.56, 2.31) | 1.09 (0.64, 1.85) |
| Sitabicha | 0.72 (0.49, 1.05) | 0.80 (0.53, 1.21) | 0.76 (0.45, 1.29) | 0.73 (0.35, 1.51) | 0.70 (0.40, 1.23) |

Abbreviations: CI, confidence interval; aHR, adjusted hazard ratio; Ref, reference.

*Regular bed net usage was defined as a person averaging > 5 nights a week sleeping under a bed net.

Significant estimates are bolded.

(aHR: 1.64, 95% CI: 1.40 to 1.94), 6 months (aHR: 1.38, 95% CI: 1.20 to 1.58), 12 months (aHR: 1.12, 95% CI: 1.00 to 1.25), or 29 months (aHR: 1.11, 95% CI: 1.01 to 1.22) (*Table 2*, *Figure 3B*). In the 29-month analysis, this relationship was modified by participant age (p-value < 0.001 by log-likelihood ratio test) with the strongest relationship between asymptomatic infection and future symptomatic malaria in children < 5 years (aHR: 1.38, 95% CI: 1.05 to 1.81), second-strongest in children 5 to 15 years (aHR: 1.16, 95% CI: 1.02 to 1.32), and weakest in adults > 15 years (aHR: 0.96, 95% CI: 0.81 to 1.13) (*Table 3*, *Figure 3—figure supplement 4*). Consistent with the 1-month analysis, we observed modification by sex in some models, with females having higher risk for symptomatic disease (*Table 3*). The limited association between asymptomatic infection and malaria over the 29-month period was also observed when using both the 'permissive' (aHR: 1.20, 95% CI: 1.11 to 1.31) and the 'stringent' (aHR 1.02, 95% CI: 0.92 to 1.13) alternate case definitions for symptomatic malaria (*Figure 3—figure supplement 4*).

We assessed for effect modification of the main exposure-outcome relationship by sex and by age in the main and pre-symptomatic models over various periods of follow-up (*Table 3*). For age, we observed only significant effect modification over the 29-month period, for which the risk of symptomatic malaria following asymptomatic infection was elevated in under-5s (aHR 1.38, 95% CI: 1.05 to 1.81) but not in adults (aHR 0.96, 95% CI: 0.81 to 1.13; p < 0.001 by log-likelihood ratio test). Conversely, for sex, we observed effect modification only at short follow-up period after an asymptomatic infection: over 1 month, the risk of symptomatic infection was lower in males (aHR 1.76, 95% CI: 1.24 to 2.50) than females (aHR 3.71, 95% CI: 2.62 to 5.24; p = 0.006 by log-likelihood ratio test).

## Short-term effect of detectability of asymptomatic infections

Owing to the consistently elevated short-term risk of symptomatic malaria in people with asymptomatic infections, we investigated the effect of parasite density in these infections on the risk of subsequent symptomatic malaria within 1 month. Compared to uninfected people, the 1-month hazard of symptomatic malaria was significantly increased by asymptomatic infections of all parasite densities,

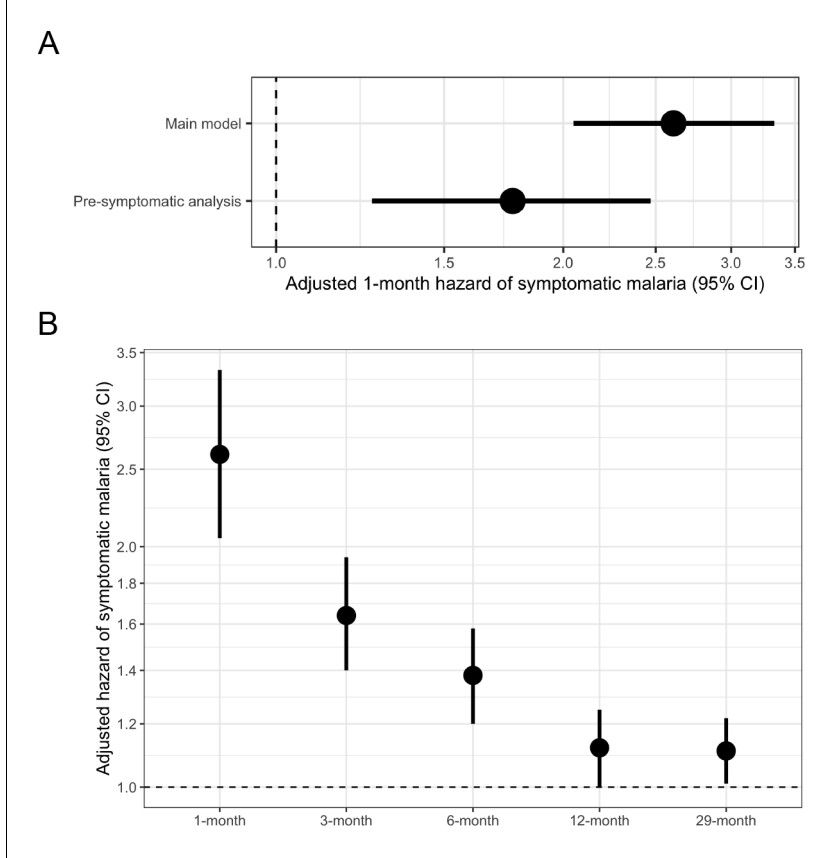

**Figure 3.** Adjusted hazard of symptomatic malaria after asymptomatic infections compared to uninfected over various follow-up periods. (**A**) Frailty Cox proportional hazards model results comparing having asymptomatic malaria infections versus being uninfected over time and the 1-month hazard of symptomatic malaria. The main model used all eligible participants while the pre-symptomatic model removed monthly follow-up visits that occurred within 14 days prior to a symptomatic malaria infection. Models controlled for covariates participant age, sex, bed net usage, and village. (**B**) Main model results using primary outcome coding of symptomatic malaria were computed using differing follow-up periods ranging from 1 to 29 months and controlled for covariates participant age, sex, bed net usage, and village.

The online version of this article includes the following figure supplement(s) for figure 3:

**Figure supplement 1.** Directed acyclic graph illustrating covariate relationships for the association between exposure to asymptomatic malaria versus no infection and time to symptomatic malaria infection.

**Figure supplement 2.** Distribution of ages (in years) of participants included in the study.

**Figure supplement 3.** Frailty Cox proportional hazards model results comparing exposure to asymptomatic malaria infections versus no infection over time and 1-month adjusted hazard of symptomatic malaria across the three case definitions for symptomatic malaria: primary, secondary permissive, and secondary stringent.

**Figure supplement 4.** Frailty Cox proportional hazards model results comparing exposure to asymptomatic malaria infections versus no infection over time and 29-month adjusted hazard of symptomatic malaria across the three case definitions for symptomatic malaria: primary, secondary permissive, and secondary stringent.

---

with the highest risk for those with densities > 1000 parasites/µL (aHR 3.99, 95% CI: 2.41 to 6.62) (*Figure 4*). This observed increase in the hazard of symptomatic malaria with increasing parasite density was most pronounced among adults >15 years (*Figure 4—figure supplement 1*); however, children's likelihood of symptomatic infection did not appear to be influenced by parasite density.

## Discussion

Using a 29-month longitudinal cohort in a high malaria transmission region of Kenya, we investigated the association between asymptomatic *P. falciparum* infections and the risk of symptomatic malaria. In the short term, compared to uninfected individuals, people of any age with asymptomatic

**Table 3.** Age- and sex-stratified adjusted hazard ratios of time to symptomatic malaria.

| Comparison | Age aHR (95% CI) | | | Sex aHR (95% CI) | |
| --- | --- | --- | --- | --- | --- |
| | **<5 years** | **5 to 15 years** | **>15 years** | **Male** | **Female** |
| 1-Month main model | 3.77 (2.02,7.04) | 2.45 (1.79,3.35) | 2.55 (1.57,4.15) | **1.76 (1.24,2.50)** | **3.71 (2.62,5.24)** |
| 1-Month pre-symptomatic | 2.85 (1.19,6.79) | 1.61 (1.05,2.46) | 1.90 (0.93,3.86) | 1.24 (0.75,2.05) | 2.34 (1.47,3.71) |
| 3-Month main model | 2.47 (1.59,3.84) | 1.49 (1.21,1.85) | 1.69 (1.23,2.32) | **1.29 (1.01,1.64)** | **2.03 (1.62,2.55)** |
| 3-Month pre-symptomatic | 2.00 (1.20,3.34) | 1.16 (0.90,1.48) | 1.35 (0.94,1.93) | 1.05 (0.79,1.39) | 1.52 (1.18,1.97) |
| 6-Month main model | 1.94 (1.34,2.80) | 1.32 (1.11,1.57) | 1.31 (1.01,1.70) | **1.13 (0.93,1.39)** | **1.62 (1.35,1.94)** |
| 6-Month pre-symptomatic | 1.63 (1.08,2.46) | 1.11 (0.92,1.34) | 1.10 (0.83,1.46) | 0.98 (0.78,1.22) | 1.33 (1.09,1.62) |
| 12-Month main model | Not calculated* | Not calculated* | Not calculated* | 1.10 (0.86,1.19) | 1.21 (1.05,1.41) |
| 12-Month pre-symptomatic | 1.24 (0.88,1.74) | 1.00 (0.86,1.17) | 0.85 (0.68,1.07) | 0.91 (0.77,1.09) | 1.04 (0.88,1.22) |
| 29-Month main model | **1.38 (1.05,1.81)** | **1.16 (1.02,1.32)** | **0.96 (0.81,1.13)** | 1.08 (0.94,1.24) | 1.14 (1.01,1.30) |
| 29-Month pre-symptomatic | **1.23 (0.92,1.64)** | **1.06 (0.93,1.21)** | **0.88 (0.74,1.05)** | Not calculated* | Not calculated* |

Abbreviations: CI, confidence interval; aHR, adjusted hazard ratio.

*Not calculated due to data sparsity.

Statistically significant effect measure modification by the log-likelihood ratio test is bolded.

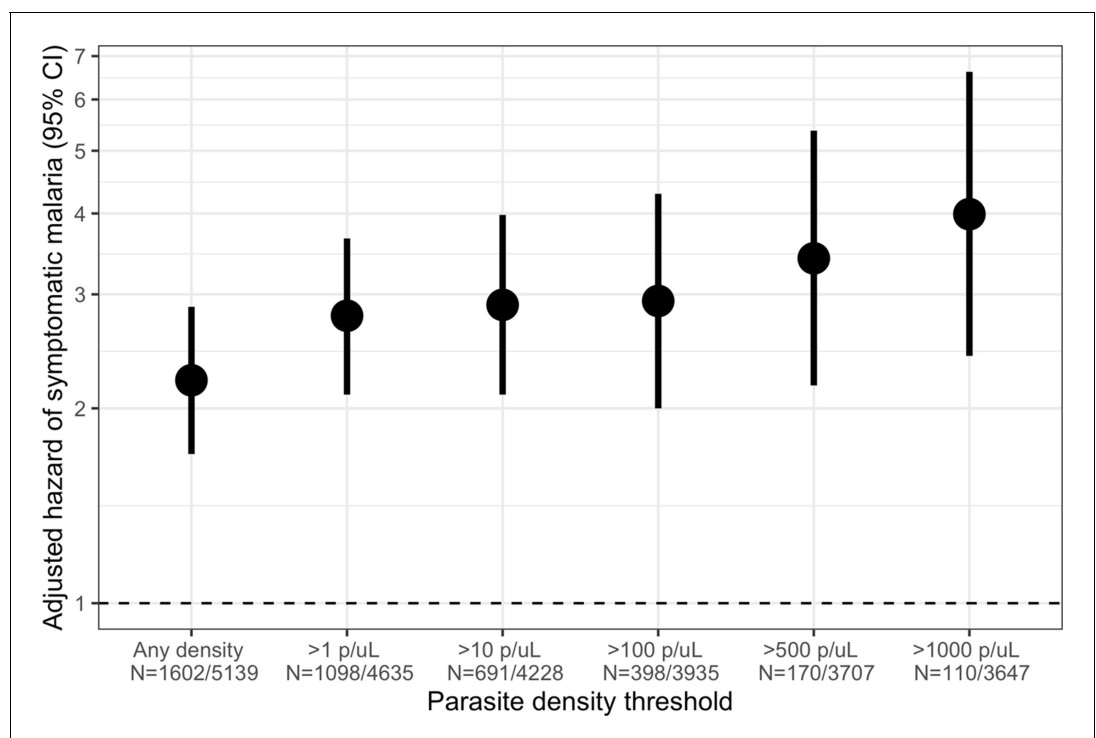

**Figure 4.** Association between parasite density of asymptomatic malaria infections and the short-term, 1-month hazard of symptomatic malaria. Estimates of the 1-month hazard of symptomatic malaria in people with asymptomatic infections compared to being uninfected are presented in separate frailty Cox proportional hazards models that were each restricted to asymptomatic infections meeting parasite density thresholds. Each model compared people with asymptomatic malaria infections meeting the listed density threshold to uninfected people and controlled for covariates participant age, sex, bed net usage, and village.

The online version of this article includes the following figure supplement(s) for figure 4:

**Figure supplement 1.** Association between parasite density of asymptomatic malaria infections and the 1-month hazard of symptomatic malaria stratified by participant age.

infections were associated with a more than twofold increased hazard of symptomatic malaria within 1 month. This elevated likelihood of symptomatic malaria was associated with asymptomatic infections at all parasite densities. As follow-up time was expanded, the association between asymptomatic infection and the increased risk of subsequent symptomatic malaria remained significant but attenuated. Collectively, our findings that detection of an asymptomatic *P. falciparum* infection confers an elevated risk of future symptomatic malaria supports the routine treatment of infections even in the absence of symptoms to prevent clinical cases.

Previous studies that detected asymptomatic infections using microscopy also reported an increased short-term hazard of symptomatic illness among children within 9 to 30 days after having an asymptomatic malaria infection (*Le Port et al., 2008*; *Njama-Meya et al., 2004*). We built upon these studies by detecting asymptomatic infections using qPCR, a highly sensitive method with a low limit of detection (*Taylor et al., 2019*), in participants of all ages and similarly found that asymptomatic infections have a high probability of being quickly followed by symptomatic illness. The increased short-term hazard could reflect misclassification of a 'pre-symptomatic' infection that progressed to symptoms as an asymptomatic exposure (*Njama-Meya et al., 2004*). This interpretation is partially supported by the diminished risk observed in a sub-analysis censoring asymptomatic infections that occurred within 14 days prior to a symptomatic event (*Figure 3A*). The increased hazard of symptomatic malaria could also have been due to the presence of new genotypes in infections (i.e. superinfection), although we previously reported that such newly apparent genotypes were associated with symptoms only in previously uninfected people (*Sumner et al., 2021*). It is notable that the increased risk of symptomatic malaria following asymptomatic infection was observed in all age groups (*Figure 3—figure supplement 2*): though children under 5 years were consistently at highest risk, the increased risk in those >15 years was surprising given the presumption that adults develop functional immunity to clinical disease possibly in part from asymptomatic carriage. Our results indicate that asymptomatic infection is associated with an increased short-term risk of malaria irrespective of age.

Elevated risk of symptomatic malaria within 1 month was present for asymptomatic infections of any parasite density. We did observe a dose-response of the risk of symptomatic malaria as a function of parasite density, particularly among adults, but the risk of malaria was increased relative to uninfected people even when they harbored low-density infections. Prior studies observed conflicting relationships between asymptomatic parasite density and subsequent malaria: PCR-positive infections below the limit of detection of microscopy were not associated with subsequent symptomatic malaria in Ugandan children (*Nsobya et al., 2004*), but were associated with a reduced risk in Malawian children (*Buchwald et al., 2019*). To our knowledge, our data are the first to analyze a broad range of parasite densities in asymptomatic infections and their association with subsequent symptomatic malaria. Though higher densities were associated with slightly higher risk of symptomatic illness, the elevated risk across all clinically detectable parasitemia does not clearly support risk stratification by detectability for the purposes of preventing clinical disease. Our results indicate that in a high-transmission setting, despite the absence of symptoms, the detection of *P. falciparum* parasites of any density is significantly associated with an increased risk of malaria in the forthcoming month and suggest that detection modality should not influence a decision to treat.

We observed significant modification of the relationship between asymptomatic infection and symptomatic malaria risk by sex. Specifically, despite an overall lower burden of symptomatic malaria among females that is consistent with prior studies (*Houngbedji et al., 2015*; *Mulu et al., 2013*; *Newell et al., 2016*), we observed that the short-term hazard of symptomatic malaria following an asymptomatic infection was significantly higher among females (aHR 3.71) compared to males (aHR 1.76). To our knowledge, this effect modification by sex has not been previously reported, with prior studies typically including sex as a covariate in models. A recent study (*Briggs et al., 2020*) highlighted large gaps in knowledge related to sex-based differences in malaria epidemiology, while reporting that Ugandan females of all ages cleared asymptomatic infections at nearly twice the rate of males. That observation, while not directly comparable, is a challenge to reconcile with ours, which suggests that asymptomatic infections in females, compared to males, are more likely to culminate in symptomatic malaria than in natural clearance. Despite similarities in cohort membership, follow-up, and outcome assessment, a key difference may be the far higher transmission intensity in our cohort: the recent application of control measures in Uganda reduced the incidence of malaria in the area by more than 10-fold and the prevalence of PCR-detectable infections more than

threefold. Either the recent fall or the current low-transmission intensity may have differentially affected the natural history of infections in that region. Reconciling these sex-based findings and exploring additional impacts relevant to prevention and control will require rigorous assessments of sex as an effect measure modifier in malaria epidemiology.

We observed that the increased risk for symptomatic disease associated with asymptomatic infection weakened as the follow-up length extended from 3 to 29 months, illustrated in both the multi-level models and Kaplan-Meier curves. One possible explanation for this observation could be inherent to the methodology whereby the magnitude of the average hazard ratio decreases as follow-up time increases (*Hernán, 2010*). Alternatively, it is biologically plausible that the further removed in time an asymptomatic exposure is, the weaker the relationship to disease outcomes, possibly by waning immunity. This is supported by our observation that older children and adults are no longer at increased risk of symptomatic disease by 29 months, although small children still maintain significantly elevated risk even for this extended follow-up period.

We used a novel approach to capture how asymptomatic malaria varied over time. Most previous work used an intention-to-treat approach for asymptomatic infections identified in cross-sectional surveys (*Henning et al., 2004*; *Liljander et al., 2011*; *Males et al., 2008*; *Nsobya et al., 2004*; *Portugal et al., 2017*; *Sondén et al., 2015*; *Wamae et al., 2019*); however, this method can misclassify person-time if the exposure frequently changes, as happens with asymptomatic infections in high-transmission areas. For previous studies with more frequent asymptomatic sampling, the projects had short follow-up periods (9 to 30 days) (*Le Port et al., 2008*; *Njama-Meya et al., 2004*), or coded the exposure as static after an asymptomatic infection occurred (*Buchwald et al., 2019*). We recorded asymptomatic malaria exposure using a time-varying method proposed by *Hernán et al., 2005* that allows participants to change exposure status throughout follow-up, which may capture a more complete view of infection dynamics with lower risk of exposure misclassification. By producing an effect estimate predictive of future risk regardless of prior exposure, this method is less prone to left truncation bias, which can occur with methods that create additive measures of months of exposure. To our knowledge, though this method (*Hernán et al., 2005*) has been used in studies of cardiovascular or kidney disease (*Danaei et al., 2013*; *Hernán et al., 2008*; *Secora et al., 2020*), it has not before been used to study malaria. Given the frequency of outcome events in high-transmission settings and the complexity of risk factors for them, this approach could be a useful addition to the analytic toolkit to assess time-varying exposures and their association with malaria outcomes.

This study had some limitations that should be considered when weighing the findings. First, asymptomatic infections were only captured at monthly follow-up visits, potentially missing transient asymptomatic infections between visits. By allowing participant exposure to vary over time, we assumed exchangeability between the exposed and unexposed groups. This was mitigated by the observation that approximately 94% of the study population changed exposure status at least once during follow-up. Finally, we estimated parasite densities using molecular methods and only at a single time point, though densities are known to fluctuate during infections. However, this potential bias in density estimations should be random and non-directional, and therefore mitigated by the analysis of over 1600 density measurements in asymptomatic infections.

In conclusion, using a novel exposure coding method and frequent sampling of both children and adults over 29 months, we found that asymptomatic *P. falciparum* infections were associated with a high likelihood of being shortly followed by symptomatic illness across all ages and parasite densities. These results suggest interventions focus on treating and reducing asymptomatic malaria in high-transmission settings.

## Acknowledgements

We are very appreciative of the Webuye study participants for their participation in this study. We also thank the project manager and field technicians in Kenya for their fastidious work: J Kipkoech Kirui, I Khaoya, L Marango, E Mukeli, E Nalianya, J Namae, L Nukewa, E Wamalwa, and A Wekesa. We thank M Emch (of the University of North Carolina at Chapel Hill) for his analysis considerations and A Nantume (of Duke University) for laboratory sample processing. This work was supported by NIAID (R21AI126024 to WPO and R01AI146849 to WPO and SMT).

## Additional information

### Funding

| Funder | Grant reference number | Author |
|---|---|---|
| National Institute of Allergy and Infectious Diseases | R21AI126024 | Wendy Prudhomme-O'Meara |
| National Institute of Allergy and Infectious Diseases | R01AI146849 | Wendy Prudhomme-O'Meara Steve M Taylor |

The funders had no role in study design, data collection and interpretation, or the decision to submit the work for publication.

### Author contributions

Kelsey M Sumner, Conceptualization, Data curation, Formal analysis, Investigation, Methodology, Writing - original draft, Writing - review and editing; Judith N Mangeni, Data curation, Investigation, Project administration, Writing - review and editing; Andrew A Obala, Conceptualization, Supervision, Investigation, Methodology, Project administration, Writing - review and editing; Elizabeth Freedman, Investigation, Methodology, Writing - review and editing; Lucy Abel, Supervision, Investigation, Project administration, Writing - review and editing; Steven R Meshnick, Conceptualization, Supervision; Jessie K Edwards, Formal analysis, Supervision, Writing - review and editing; Brian W Pence, Supervision, Methodology, Writing - review and editing; Wendy Prudhomme-O'Meara, Conceptualization, Resources, Supervision, Funding acquisition, Methodology, Writing - review and editing; Steve M Taylor, Conceptualization, Resources, Supervision, Funding acquisition, Investigation, Methodology, Project administration, Writing - review and editing

### Author ORCIDs

Steve M Taylor ⓘ https://orcid.org/0000-0002-2783-0990

### Ethics

Human subjects: The study was approved by institutional review boards of Moi University (2017/36), Duke University (Pro00082000), and the University of North Carolina at Chapel Hill (19-1273). All participants or guardians provided written informed consent, and those over age 8 provided additional assent.

### Decision letter and Author response

Decision letter https://doi.org/10.7554/eLife.68812.sa1
Author response https://doi.org/10.7554/eLife.68812.sa2

## Additional files

### Supplementary files

• Supplementary file 1. Comparison of time-varying exposure coding approaches.

• Supplementary file 2. Covariate distribution across symptomatic events: secondary permissive case definition.

• Supplementary file 3. Covariate distribution across symptomatic events: secondary stringent case definition.

• Reporting standard 1. STROBE checklist for cohort studies.

• Transparent reporting form

### Data availability

Data will be shared under the auspices of the Principal Investigators. Investigators and potential collaborators interested in the datasets will be asked to submit a brief concept note and analysis plan.

Requests will be vetted by Drs. O'Meara and Taylor and appropriate datasets will be provided through a password protected secure FTPS link. No personal identifying information will be made available to any investigator. Relevant GPS coordinates would only be provided when (1) the planned analysis cannot reasonably be accomplished without them and (2) the release of the coordinates is approved by the Institutional Review Board. A random error in the latitude and longitude of 50-100 meters will be added to each pair of coordinates to protect individual household identities. General de-identified datasets will be prepared that can accommodate the majority of requests. These will be prepared, with documentation, as the data is cleaned for analysis in order to reduce time and resources required to respond to individual requests. Recipients of study data will be asked to sign a data sharing agreement that specifies what the data may be used for (specific analyses), criteria for acknowledging the source of the data, and the conditions for publication. It will also stipulate that the recipient may not share the data with other investigators. Requests for data use must be made directly to the PI and not through third parties.

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
