## [Decision Letter]

**Acceptance summary:**

This study addresses an important topic with major public health implications: whether chronic, asymptomatic parasite carriage is associated with increased or decreased risk of subsequent episodes of clinical malaria in Africa. The authors followed up individuals exposed to intense malaria transmission in western Kenya. They used a novel analytical approach to assess the association between baseline asymptomatic infections and subsequent symptomatic malaria and found that the risk of disease is higher if someone carried asymptomatic *Plasmodium falciparum* infection at the baseline, compared to uninfected individuals.

**Decision letter after peer review:**

Thank you for submitting your article "Impact of asymptomatic *Plasmodium falciparum* infection on the risk of subsequent symptomatic malaria in a longitudinal cohort in Kenya" for consideration by *eLife*. Your article has been reviewed by 3 peer reviewers, including Marcelo U Ferreira as the Reviewing Editor and Reviewer #1, and the evaluation has been overseen by a Senior Editor. The following individual involved in review of your submission has agreed to reveal their identity: Cristian Koepfli (Reviewer #2).

Essential revisions:

1. How was the cohort assembled to support the representativeness of the participants for the general community?

2. Which symptoms were asked about in the self-report, and what was the algorithm used to define a case as symptomatic or asymptomatic, over what time period?

3. Was any participant classified as symptomatically infected at a monthly visit? If so what status were they given at that visit?

4. How many participants were symptomatic, PCR positive but RDT negative, and therefore not treated? How did their clinical course, if any, compare with those who were also RDT positive and therefore treated?

5. Was there any correlation by individuals in the same household?

6. Are you able to control for spatial heterogeneity in risk, that could also help tease out whether superinfection has confounded your results?

7. Please reconsider your suggestion of causality between asymptomatic and symptomatic infections.

8. Were the time periods that occurred after an individual was treated for a symptomatic illness different than before they had an illness? I noted that treatment was not included in the models.

9. Given that some parasite genotyping data are already available for this cohort, could the authors incorporate genotyping data in their analysis to distinguish between persisting infections that became symptomatic vs. superinfection with new strains?

10. The authors ascertain the exposure status at every monthly follow-up visit, thereby treating each of these visits as a new study entry. It remains unclear whether this method is applicable for infectious diseases for which the risk of subsequent infections changes over time. The key question is whether this change is moderate or not over the time period considered. The stratification by age shows substantial changes over five year periods which is not that much more than a 29-month follow up. Please discuss whether these results are robust against the deviations from moderate time dependence.

*Reviewer #1:*

The "premunition" hypothesis postulares that individuals living in malaria-endemic areas gradually develop a form of clinical immunity characterized by the chronic carriage of low parasite burdens that reduces the risk of superinfection with more virulent strains and protects from full-blown disease (e.g., doi: 10.1590/s0074-02761994000600013). Whether this hypothesis holds in African settings with contrasting levels of transmission intensity remains unknown.

Here, Sumner and colleagues provide evidence that, contrary to the premunition hypothesis, pre-existing asymptomatic infections at the baseline are associated with increased (rather than decreased) short-term risk of clinical disease in a cohort of 268 Kenyans exposed to intense malaria transmission and followed up for 29 months. This finding has major public health implications as it implies that treating asymptomatic infections (that are typically missed by malaria control strategies) may reduce subsequent malaria-related morbidity.

Sumner and colleagues use multivariate frailty Cox proportional hazards models to account for the repeated observations per individual – an interesting approach that has not been previously applied (to my knowledge) to the study of infectious diseases. One possible caveat is that the model does not account/correct for some degree of acquired immunity elicited by a previous infection diagnosed during the follow-up. For example, if the study participant is uninfected at baseline, asymptomatically infected during the first follow-up visit and again uninfected during the second follow-up visit, his/her level of "clinical immunity" (and thus the odds of having clinical disease) may have changed from the baseline (uninfected) to the second visit (currently uninfected, but recently exposed to the parasite). This might be discussed in the manuscript.

The key biological question underlying this study is whether chronically infected but asymptomatic individuals are more likely than their uninfected counterparts to develop clinical disease because pre-existing parasites eventually multiply and parasite density crosses the (individual) "fever threshold", causing malaria-related symptoms, or, alternatively, whether these infected hosts are more likely (e.g., because of increased exposure) to be superinfected with more virulent parasite strains over the next few weeks and therefore develop clinical disease. Surprisingly, this question remains unaddressed in the present manuscript.

However, the same authors have recently reported a detailed molecular analysis of incident *P. falciparum* infections in this cohort (doi: 10.1093/cid/ciab357). Molecular data suggest that the second hypothesis is correct: incident infections with only novel haplotypes were significantly associated with increased odds of subsequent symptomatic malaria over 14 months of follow-up (doi: 10.1093/cid/ciab357). Integrating these separately reported results in a single publication would render it much more appealing for the broad audience of *eLife* readers.

In conclusion, Sumner et al. convincingly show that asymptomatic parasite carriage in a high-transmission setting in Kenya is associated with increased subsequent risk of clinical malaria, but the mechanisms underlying this association remain relatively unexplored in this study.

Comments for the authors:

This is a clearly written manuscript that addresses a topic of major public health significance. My only concern refers to the way the interesting results reported here are explored.

The authors use a relatively new approach to survival analysis to show that carriers of asymptomatic malaria infections are more likely than uninfected individuals in the same populations to develop a clinical disease over the next month, although not necessarily over longer periods of follow-up. To this end, they excluded putatively "pre-symptomatic" infections, that were arbitrarily defined as those occurring within 14 days prior to a symptomatic infection. I see no clear justification for this 14-day time window.

I would favor an alternative, more strict approach: excluding infections leading to clinical symptoms only if identical parasite haplotypes persisted since the first parasite detection until the clinical episode. This would provide evidence of "pre-symptomatic" infections. Combining changes in parasitemia over time would make the case even stronger. Quantifying the proportion of subsequent symptomatic infections associated with persisting haplotypes would provide an information of major biological and epidemiological interest and would allow us to test whether the arbitrary time window of 14 days is appropriate to rule out "pre-symptomatic infections". The authors have already genotyped a large proportion of incident infections in cohort participants (doi: 10.1093/cid/ciab357) but do not consider this rich dataset in their analyses.

The exposure status of each study participant (whether asymptomatically infected or not) was ascertained at every monthly follow-up visit and allowed to vary each month. This is an interesting approach that has been previously used in epidemiological studies of non-communicable diseases, but not of infections. The caveat here is that each new infection (either persisting or self-cured) may reduce the subsequent risk of malaria-related disease. Therefore, someone who is uninfected (at baseline), then infected (first follow-up visit) and again uninfected (second follow-up visit) has a similar "exposure status" during the second follow-up visit as someone who remained uninfected since the baseline evaluation, but this (mis)classification ignores acquired immunity gradually developed in response to each infection experienced by cohort participants, which may reduce the subsequent risk of infection and disease.

Not surprisingly, the risk of disease following asymptomatic parasite carriage increases with increasing parasite density (Figure 4), but I wonder whether age-related fever thresholds may be observed in this population. Given that parasite density data are available, I suggest that a further analysis of parasite density levels associated with symptoms across age groups might provide some interesting insights into the biological/immunological mechanisms underlying the main study finding.

*Reviewer #2:*

1) At least three processes could potentially result in more frequent clinical infection following asymptomatic infection. First, the infection might be presymptomatic. Each clinical infection is preceded by a short period (a few days) of asymptomatic blood-stage infection. Second, chronic asymptomatic infections might increase in density though hitherto unknown processed, and cause clinical illness. Third, asymptomatic infection might be an indicator for higher exposure. Subsequent new infections might then result in clinical disease.

Depending on the scenario, different control interventions are warranted. If asymptomatic infections develop into symptomatic infections, treating these infections will reduce the number of clinical cases. If they are a marker for higher exposure, vector control will be warranted.

The first option was analyzed by a subgroup analysis (asymptomatic infections <14 days before clinical disease excluded from analysis). The results showed a surprisingly large difference between models, indicating that a large number of 'asymptomatic' infections was in fast presymptomatic.

Scenario 2 and 3 could be distinguished by parasite genotyping, as a new infections will most likely be a different clone. Thus, genotyping would add substantially to the interpretation of the data. Given that the authors already have published genotyping results from a similar study (Sumner et al., 2021), this should be straightforward. It might also help to understand better why some previous studies have found similar results as this one, while other studies have found contrasting results.

2) Seasonality was not considered in the analysis, but might be important to understand the observations. For example, the finding that "a difference in the time to symptomatic malaria in the first few months post asymptomatic infection but not long-term" could be confounded by the fact that most transmission occurs in the wet season, and thus that the probability for both asymptomatic and clinical infection is higher in the wet season.

3) Lines 52-54: It would be good to mention whether these estimates are based on microscopy or PCR. It would also be good to introduce the concept of supatent infections in the introduction, and discuss the possibility that even by PCR some infections were missed.

4) Lines 188-189: While I like brevity, this section should be expanded, given its importance for the study. How many participants were never infected? How often was someone infected at a single time point? What was the duration of infections? Did you observe the sequence infected-non infected-infected, pointing to possible undetected infections at certain time points (in particular if the same clone was observed)?

Likewise, given that age was an important variable, please provide more details on the age of clinical patients. A figure showing age trends in asymptomatic and clinical infections would be helpful.

*Reviewer #3:*

Sumner et al. report on a prospective cohort study of 254 participants aged 1 to 85 years living in three villages in Western Kenya. Participants gave monthly blood samples for testing by polymerase chain reaction (PCR) for parasite DNA and recorded any symptoms of illness during the previous 30 days from June 2017 through November 2019. Patients who reported symptoms of illness at any time were tested for malaria by rapid diagnostic test (RDT) and had blood samples taken for analysis by PCR. The study assessed whether asymptomatic individuals carrying malaria parasites were more likely, over the following time period, to have a symptomatic episode of malaria than people not infected with malaria. Using an analysis that treated each monthly visit as a new entry in the study, the time to symptomatic malaria or the end of the study was calculated and attributed to the status of the participant in that month (uninfected or infected but asymptomatic). Repeated measures on each participant were accounted for in the models using a random-effect measure at the level of participant.

The researchers found that participants with asymptomatic infections developed symptomatic infections more quickly than those uninfected (median time to symptomatic malaria: 173 days [interquartile range, IQR: 49, 399] vs. 230 days [IQR: 98, 402], respectively). There was a statistically significant difference in the hazard ratio comparing symptomatic malaria between asymptomatic infected and uninfected participants for periods from 1- to 6-months, but not at 12 months, after adjusting for covariates, with a decline in the hazard ratio over longer follow-up periods.

These data add to the ongoing discussion about the importance and clinical relevance of asymptomatic malaria infections. There is widespread recognition that a substantial proportion of the human malaria parasite reservoir in malaria-endemic areas is found in people who do are asymptomatic, paucisymptomatic or afebrile. There is a debate, however, as to whether these infections are: a) the result of previous infections on their way to being resolved; b) chronic infections resulting from partial immunity to malaria; or c) new onset patent parasitemia appearing prior to the onset of symptoms. Additionally, the long-term outcomes of these infections, and their potential deleterious effects, have been the subject of debate. Such infections could: a) disappear without intervention; b) remain for long periods of time without causing harm; c) remain for long periods of time causing mild intermittent symptoms; or d) produce symptoms of illness that prompt healthcare seeking and treatment.

The authors of this paper have taken great care in the design and analysis of the complex data gathered during this study, some aspects of which have been reported elsewhere (O'Meara et al. JID 2020 221;1176-1184). However, the authors' conclusion that asymptomatic infections should be treated with antimalarials because the asymptomatic infections themselves increase the risk of future symptomatic malaria is not fully supported by the evidence. The findings in this study could also result from superinfection, that is, if individuals with asymptomatic infections are more likely to be infected again, a new parasite strain could provoke symptoms of illness. If an asymptomatic infection serves as a marker of living in an area with a higher force of infection, then it would follow that such individuals are at greater risk of multiple infections. The development of symptoms associated with a new infection by a different parasite strain would, therefore, not be affected by treatment of a previous asymptomatic infection.

There is already some suggestion in the report that symptomatic infections were associated with a higher force of infection. The study reported that time to symptomatic malaria was shorter for participants living in the village of Maruti. A previous paper reporting on entomologic indices in the study area showed that this village had the highest density of malaria mosquito vectors per household, and effectively the highest entomologic inoculation rate, compared to the other villages (O'Meara et al. JID 2020 221;1176-1184). Additionally, the previous paper reported that 41 individuals had more than 1 symptomatic infection, suggesting heterogeneity in transmission and force of infection.

Additionally, it was surprising to see that age, which is generally a proxy for greater lifetime exposure to malaria and therefore a higher degree of partial immunity, did not modify the association between asymptomatic infection and subsequent symptomatic infection. This finding could also be explained by the symptomatic infections arising from reinfection with new strains of parasites more likely to cause symptoms.

The strengths of the report lie in the use of a novel approach to time-varying exposure status (asymptomatic infection vs. uninfected) that avoided misclassification of person-time; careful statistical analysis of the data, including excluding post-baseline potential pre-symptomatic episodes; and relatively complete follow-up of a modest cohort of participants.

The weaknesses in this paper, however, limit the paper's ability to conclude that asymptomatic infections should be treated to avoid symptomatic illness. In addition to the points mentioned above, the analysis did not control for the timing of episodes. Although not reported in this paper, the previous paper by this group suggests that the proportion of the cohort that had asymptomatic malaria infections at baseline was >50%. These baseline infections appear to be counted as asymptomatic infections although the length of time that these individuals had been infected was unknown. The authors could clarify whether these baseline asymptomatic infections were more likely to become symptomatic than asymptomatic infections that developed later, perhaps suggesting that many of them were presymptomatic.

Comments for the authors:

The paper is very well written and conceived, but I do not agree that the main conclusion is supported by the data.

Comments on the statistical analysis:

1. Line 105-107: missed monthly visits got an exposure status equal to the previous month's value – was any sensitivity analysis done?

2. Line 107-109: why are people considered lost to follow-up and censored at the time of the imputed monthly visit; I would argue imputation is best not done in this situation

3. Line 136-138: How do the authors account for repeated measures using a Bonferroni correction?

4. Line 143-144; Equation 1: What is ϵi in this formula? What are the underlying distributional assumptions for αi and were these assumptions challenged? Please clarify notation used.

5. Individual are clustered in households (38 in total); was this taken into account in the analysis? This would be very relevant for malaria. A nested frailty approach would be useful here.

6. Throughout the text it would help to clarify which HR are referred to: conditional or unconditional (wrt the frailty). This can be briefly mentioned so that repetition is not needed.

---

## [Author Response]

Essential revisions:1. How was the cohort assembled to support the representativeness of the participants for the general community?

The cohort was assembled using radial sampling of 12 households per village for three villages within Webuye, Western Kenya. The first household in each village was randomly selected. The three villages were geographically close and chosen due to their high malaria prevalence in a previous cross-sectional study in the area [Mean *P. falciparum* prevalence in 2013: Kinesamo (18.4%), Maruti (20.8%), Sitabicha (22.8%), Obala AA *PloS One* 2015;10(7)]. We have added some text to the methods to describe this:

“The cohort was assembled using radial sampling of 12 households per village for three villages with high malaria transmission. The first household in each village was randomly selected. Two households moved during follow-up and were replaced.”

2. Which symptoms were asked about in the self-report, and what was the algorithm used to define a case as symptomatic or asymptomatic, over what time period?

We have clarified this by adding to the *Exposure and outcome ascertainment* section of the manuscript: “The main exposure was an asymptomatic *P. falciparum* infection during monthly active case detection assessments, defined as *P. falciparum*-positive by qPCR in a person lacking symptoms. People who were *P. falciparum*-negative by qPCR during monthly visits were considered uninfected. We defined symptomatic *P. falciparum* infection as the *current* presence of at least one symptom consistent with malaria during a sick visit (i.e. fever, aches, vomiting, diarrhea, chills, cough or congestion) and *P. falciparum*-positive by both RDT and qPCR.” Additional details surrounding exposure and outcome ascertainment are provided in the supplement as described in #3 below.

3. Was any participant classified as symptomatically infected at a monthly visit? If so what status were they given at that visit?

Yes, and this information is now described in the supplement under the header Exposure and outcome ascertainment:

“Some participants were classified as symptomatically infected at a monthly visit through passive detection of symptoms; this occurred if a study team member conducting a monthly visit was approached by a participant reporting malaria-like symptoms. […] If that person did not meet our case definition for symptomatic malaria, then they were removed from follow-up for that month and re-entered for follow-up in the following month.”

4. How many participants were symptomatic, PCR positive but RDT negative, and therefore not treated? How did their clinical course, if any, compare with those who were also RDT positive and therefore treated?

In the primary analysis data set, we included 0 participants who were symptomatic, PCR positive, and RDT negative, because they did not meet our case definition for symptomatic malaria. Looking back at the original data set prior to any censoring criteria being applied, there were 6016 observations collected through monthly and sick visits and 183/6016 (3.0%) of them met the criteria of being symptomatic, PCR positive, and RDT negative. Subsetting the data set to only sick visits, 183/983 (18.6%) of sick visits were symptomatic, PCR positive, and RDT negative. Because this number was fairly few across a large number of participants and 29 months of observation, we have limited ability to make inferences about the natural history of malaria in RDT-negative/PCR-positive people; however, we did conduct a secondary analysis where we computed the hazard of symptomatic malaria using a secondary permissive definition for symptomatic malaria (PCR-positive and having at least one malaria-like symptom) to assess if alternative symptomatic malaria case definitions influenced results. Using the secondary permissive case definition for symptomatic malaria, we observed overall similar results to those produced using the primary case definition. We have added this to the main text:

“We observed similar 1-month elevated risks of malaria in asymptomatically-infected people when using both the “permissive” [aHR 1.97, 95% CI 1.63 to 2.40] and the “stringent” [aHR 2.76, 95% CI 2.11 to 3.62] alternate case definitions for symptomatic malaria (Figure 3—figure supplement 3).”

5. Was there any correlation by individuals in the same household?

We chose to only include a random intercept at the individual and not household level due to the little impact correlation at the household level had on the relationship between asymptomatic/uninfected at monthly visits and subsequent symptomatic malaria. We assessed the possibility of using a nested frailty approach with random intercepts at the individual and household levels. To do so, we re-computed the frailty Cox proportional hazards model described in Equation 1 in the main text using a random intercept at the individual level and an additional random intercept at the household level. Results were similar for the 1-month hazard of symptomatic malaria for those asymptomatically-infected versus uninfected when a random intercept at the household level was included [aHR: 2.64, 95% CI: 2.07 to 3.37] versus excluded [aHR: 2.61, 95% CI: 2.05 to 3.33]. Because including a household level random intercept had little impact on model results, we excluded it from analyses to be able to have model convergence to test effect measure modification of the relationship by age and sex.

6. Are you able to control for spatial heterogeneity in risk, that could also help tease out whether superinfection has confounded your results?

We controlled for spatial heterogeneity in risk by including village fixed effects in the models. We do not expect spatial risk to be substantially different within and between villages due to the radial sampling of households, close proximity of villages to each other (all less than 11 km apart), and similar high malaria prevalence across the three villages [Mean *P. falciparum* prevalence in 2013: Kinesamo (18.4%), Maruti (20.8%), Sitabicha (22.8%), Obala AA *PloS One* 2015;10(7)].

7. Please reconsider your suggestion of causality between asymptomatic and symptomatic infections.

We have relaxed the language in the main text to focus more on the association between asymptomatic and symptomatic infections instead of a causal relationship. To do so, we changed the casual language. For example, the Abstract now reads “Compared to being uninfected, asymptomatic infections were associated with an increased 1-month likelihood of symptomatic malaria [adjusted Hazard Ratio (HR):2.61, 95%CI:2.05–3.33]…” We made similar changes in the Introduction, Results, and Discussion, each of which are tracked in the accompanying documents.

8. Were the time periods that occurred after an individual was treated for a symptomatic illness different than before they had an illness? I noted that treatment was not included in the models.

To test if prior treatment during the study period influenced results, we re-computed the frailty Cox proportional hazards model in Equation 1 with a covariate for prior treatment. To do so, we reran the frailty Cox proportional hazards model in Equation 1 with an additional covariate representing prior receipt of antimalarial treatment. This variable was coded dichotomously as having received study-prescribed antimalarials up until that monthly visit or not. For example, a person was coded as having not received study-prescribed antimalarials up until their first symptomatic infection, but afterward were coded as receiving treatment from that point forward in follow-up. Compared to primary models, antimalarial treatment had minimal effect on the hazard of symptomatic malaria when asymptomatically infected versus uninfected in the short-term [1-month aHR: 2.61, 95% CI: 2.05 to 3.33] but made results largely null in the long-term [29-month aHR: 1.04, 95% CI: 0.95 to 1.15].

We have added to the methods section Sensitivity analyses:

“For prior antimalarial treatment, we included a variable coded dichotomously as having received study-prescribed antimalarials up until that monthly visit or not; a person was coded as having not received study-prescribed antimalarials up until their first symptomatic infection, but afterward were coded as receiving treatment from that point forward in follow-up,” and the Results the general statement that “This association was similar in a model adjusted for covariates [adjusted HR (aHR): 2.61, 95% CI: 2.05 to 3.33] (Table 2, Figure 3A) as well as when using alternative modeling approaches, alternate outcome case definitions, and in sensitivity analyses.”

9. Given that some parasite genotyping data are already available for this cohort, could the authors incorporate genotyping data in their analysis to distinguish between persisting infections that became symptomatic vs. superinfection with new strains?

Unfortunately, parasite genotype data are unavailable for more than half of the months of observation that we analyze here. Our recent analysis of parasite genotypes used data from the first phase of this cohort [Sumner KM *Clin Infect Dis* 2021;ciab357], and we supplemented these 14 months of phase 1 with an additional 15 months from phase 2 for this analysis.

Results from that paper [Sumner KM *Clin Infect Dis* 2021;ciab357] suggested an association between the presence of new haplotypes in incident infections and an increased likelihood of symptomatic malaria; however, this relationship was overall null for persistent infections. We have referenced some of these results into the discussion:

“The increased short-term hazard could reflect misclassification of a “pre-symptomatic” infection that progressed to symptoms as an asymptomatic exposure (Njama-Meya et al., 2004). […] The increased hazard of symptomatic malaria could also have been due to the presence of new genotypes in infections (i.e. superinfection), although we previously reported that such newly-apparent genotypes were associated with symptoms only in previously-uninfected people (Sumner et al., 2021).”

10. The authors ascertain the exposure status at every monthly follow-up visit, thereby treating each of these visits as a new study entry. It remains unclear whether this method is applicable for infectious diseases for which the risk of subsequent infections changes over time. The key question is whether this change is moderate or not over the time period considered. The stratification by age shows substantial changes over five year periods which is not that much more than a 29-month follow up. Please discuss whether these results are robust against the deviations from moderate time dependence.

To account for the gradual acquisition of immunity in cohort participants and subsequently time dependence, we have re-computed the models including a variable for the number of prior asymptomatic infections a person experienced in the study up until each monthly visit. To assess how the number of prior asymptomatic infections a person had during the study period could have influenced results, we re-computed the frailty Cox proportional hazards model in Equation 1 with a variable added for the number of prior asymptomatic infections each person had in the study up until each monthly follow-up visit. The covariate for number of prior infections was included as a continuous number in the model. We observed that the number of prior asymptomatic infections had little impact on the hazard of symptomatic malaria comparing those asymptomatically infected to uninfected in both the short-term [1-month aHR: 2.60, 95% CI: 2.03 to 3.31] and long-term [29-month aHR: 1.29, 95% CI: 1.17 to 1.42] compared to the primary models” Overall, we did not observe a large difference in the short-term or long-term results when accounting for the number of prior infections a person experienced, suggesting that the results were robust to deviations from time dependence.

We have added to the methods section Sensitivity analyses:

“For the number of prior infections, we included in the model as a covariate the number of prior infections as a continuous number,” and to the Results the general statement that “This association was similar in a model adjusted for covariates [adjusted HR (aHR): 2.61, 95% CI: 2.05 to 3.33] (Table 2, Figure 3A) as well as when using alternative modeling approaches, alternate outcome case definitions, and in sensitivity analyses.”

Reviewer #1:[…] Comments for the authors:This is a clearly written manuscript that addresses a topic of major public health significance. My only concern refers to the way the interesting results reported here are explored.The authors use a relatively new approach to survival analysis to show that carriers of asymptomatic malaria infections are more likely than uninfected individuals in the same populations to develop a clinical disease over the next month, although not necessarily over longer periods of follow-up. To this end, they excluded putatively "pre-symptomatic" infections, that were arbitrarily defined as those occurring within 14 days prior to a symptomatic infection. I see no clear justification for this 14-day time window.I would favor an alternative, more strict approach: excluding infections leading to clinical symptoms only if identical parasite haplotypes persisted since the first parasite detection until the clinical episode. This would provide evidence of "pre-symptomatic" infections. Combining changes in parasitemia over time would make the case even stronger. Quantifying the proportion of subsequent symptomatic infections associated with persisting haplotypes would provide an information of major biological and epidemiological interest and would allow us to test whether the arbitrary time window of 14 days is appropriate to rule out "pre-symptomatic infections". The authors have already genotyped a large proportion of incident infections in cohort participants (doi: 10.1093/cid/ciab357) but do not consider this rich dataset in their analyses.

We have updated the manuscript text to reflect why the 14 day (2 week) time window was chosen for the pre-symptomatic subset analysis:

“The time frame for identifying potentially pre-symptomatic infections was chosen for consistency with previous work studying time to symptomatic malaria (Buchwald et al., 2019).”

Unfortunately parasite genotype data are only available for less than half of the time period of this cohort. Additionally, incorporation of these complex deep-sequenced genotype data would vastly expand the scope and impact of this report, which is focused on the clinically-actionable aspects of the natural history of asymptomatic infections.

The exposure status of each study participant (whether asymptomatically infected or not) was ascertained at every monthly follow-up visit and allowed to vary each month. This is an interesting approach that has been previously used in epidemiological studies of non-communicable diseases, but not of infections. The caveat here is that each new infection (either persisting or self-cured) may reduce the subsequent risk of malaria-related disease. Therefore, someone who is uninfected (at baseline), then infected (first follow-up visit) and again uninfected (second follow-up visit) has a similar "exposure status" during the second follow-up visit as someone who remained uninfected since the baseline evaluation, but this (mis)classification ignores acquired immunity gradually developed in response to each infection experienced by cohort participants, which may reduce the subsequent risk of infection and disease.

Addressed in Essential revisions #10 above.

Not surprisingly, the risk of disease following asymptomatic parasite carriage increases with increasing parasite density (Figure 4), but I wonder whether age-related fever thresholds may be observed in this population. Given that parasite density data are available, I suggest that a further analysis of parasite density levels associated with symptoms across age groups might provide some interesting insights into the biological/immunological mechanisms underlying the main study finding.

As suggested, we did an additional analysis assessing the 1-month hazard of symptomatic malaria comparing people with asymptomatic infections with parasite densities above a series of thresholds to people who were uninfected, stratified by people’s age groups (<5 years, 5-15 years, and >15 years). Among adults >15 years, we still saw a positive association between increasing parasite density in asymptomatic infections and a higher 1-month hazard of symptomatic malaria (see Figure 4—figure supplement 1); however, parasite density had less of an effect among children. We have added this analysis to the main text:

“As an additional analysis, we repeated this process for each parasite density threshold stratified by participant age (<5 years, 5-15 years, >15 years). This observed increase in the hazard of symptomatic malaria with increasing parasite density was most pronounced among adults >15 years (Figure 4—figure supplement 1); however, children’s likelihood of symptomatic infection did not appear to be influenced by parasite density.”

Reviewer #2:1) Seasonality was not considered in the analysis, but might be important to understand the observations. For example, the finding that "a difference in the time to symptomatic malaria in the first few months post asymptomatic infection but not long-term" could be confounded by the fact that most transmission occurs in the wet season, and thus that the probability for both asymptomatic and clinical infection is higher in the wet season.

To investigate the effect of seasonality on our results, we re-computed the models including seasonality as a covariate. To investigate how seasonality could have affected results, we re-computed the frailty Cox proportional hazards model in Equation 1 while including a variable for seasonality. We classified monthly visits that occurred any time from May to October as the high transmission season and from November to April as the low transmission season, based on the region’s rainy seasons. With seasonality in the model, a high hazard of symptomatic malaria was observed among those that were asymptomatically infected compared to uninfected at monthly visits [1-month aHR: 2.46, 95% CI: 1.93 to 3.15; 29-month aHR: 1.14, 95% CI: 1.04 to 1.26]; these results were similar to the main analysis presented in the manuscript using Equation 1, suggesting that the increased hazard of symptomatic malaria following asymptomatic infection was not solely attributable to seasonality.

We have added to the methods section Sensitivity analyses:

“For seasonality, we classified monthly visits that occurred any time from May to October as the high transmission season and from November to April as the low transmission season, based on the region’s rainy seasons.,” and to the Results the general statement that “This association was similar in a model adjusted for covariates [adjusted HR (aHR): 2.61, 95% CI: 2.05 to 3.33] (Table 2, Figure 3A) as well as when using alternative modeling approaches, alternate outcome case definitions, and in sensitivity analyses.”

2) Lines 52-54: It would be good to mention whether these estimates are based on microscopy or PCR. It would also be good to introduce the concept of supatent infections in the introduction, and discuss the possibility that even by PCR some infections were missed.

We have now added the specific malaria diagnostics used for the estimates:

“In 2015, a geo-spatial meta-analysis estimated a continent-wide prevalence of asymptomatic *P. falciparum* in children aged 2-10 years old of 24% based on microscopy and rapid diagnostic test results (RDT) (Snow et al., 2017).”

3) Lines 188-189: While I like brevity, this section should be expanded, given its importance for the study. How many participants were never infected? How often was someone infected at a single time point? What was the duration of infections? Did you observe the sequence infected-non infected-infected, pointing to possible undetected infections at certain time points (in particular if the same clone was observed)?

We have added some details to the first paragraph of results describing participant follow-up: “… person-months of asymptomatic malaria exposure; the median total months of asymptomatic exposure for a participant was 9 (IQR: 5, 17). Exposure status frequently changed for participants and remained constant for only 16 (6.2%) people across follow-up; 4 people were asymptomatically infected for the entirety of follow-up and only 12 people were never infected (Figure 2A).”

Likewise, given that age was an important variable, please provide more details on the age of clinical patients. A figure showing age trends in asymptomatic and clinical infections would be helpful.

We have added more detail about the age of the participants to the main text and have added Figure 3—figure supplement 2 illustrating the age distribution: “… and a median age of 13 years (range: 1, 85) (Figure 3—figure supplement 2).”

Reviewer #3:Comments for the authors:The paper is very well written and conceived, but I do not agree that the main conclusion is supported by the data.Comments on the statistical analysis:1. Line 105-107: missed monthly visits got an exposure status equal to the previous month's value – was any sensitivity analysis done?

Our primary models impute missed monthly visits to have an exposure status equal to the previous month’s value. We chose this approach based on a previous paper that had employed this approach to study asymptomatic malaria over time [Nguyen *Lancet Infect Dis* 2018;18(5)]. Imputing missed monthly visits added 715 observations to the primary analysis data set, with imputed monthly visits making up approximately 13% of the observations in the data set (715/5379).

We have added a sensitivity analysis to the manuscript where we repeated the 1 and 29-month hazard of symptomatic malaria analyses using the original data set without imputation for missed monthly visits. We found that results produced using the data set without imputation [1-month aHR: 2.75, 95% CI: 2.05 to 3.66; 29-month aHR: 1.40, 95% CI: 1.22 to 1.61] were similar to the results presented in the manuscript where imputation was performed [1-month aHR: 2.61, 95% CI: 2.05 to 3.33; 29-month aHR: 1.11, 95% CI: 1.01 to 1.22]; notably, imputation produced estimates closer to the null of 1, suggesting that bias from imputation was towards the null.

We have added to the Exposure and outcome ascertainment section “A sensitivity analysis was conducted for imputation using a dataset without imputation for missed monthly visits,” and to the Results “This association was similar in a model adjusted for covariates [adjusted HR (aHR): 2.61, 95% CI: 2.05 to 3.33] (Table 2, Figure 3A) as well as when using alternative modeling approaches, alternate outcome case definitions, and in sensitivity analyses.”

2. Line 107-109: why are people considered lost to follow-up and censored at the time of the imputed monthly visit; I would argue imputation is best not done in this situation.

Addressed above.

3. Line 136-138: How do the authors account for repeated measures using a Bonferroni correction?

We applied the Bonferroni correction to all the table *p*-values for Table 1 by multiplying each *p*-value by 29, which was the maximum amount of follow-up visits (and repeated measures) each person could have in the study.

4. Line 143-144; Equation 1: What is ϵi in this formula? What are the underlying distributional assumptions for αi and were these assumptions challenged? Please clarify notation used.

We have clarified the definitions for these terms in the main text:

“We allowed the main exposure to vary each month based on the monthly follow-up visit infection status (m), and included a random intercept at the participant level (αi) to account for potential correlated intra-individual outcomes. A log-normal distribution was used for the random effect ϵi. represented the model’s error term.”

5. Individuals are clustered in households (38 in total); was this taken into account in the analysis? This would be very relevant for malaria. A nested frailty approach would be useful here.

Addressed in Essential revisions #5 above.

6. Throughout the text it would help to clarify which HR are referred to: conditional or unconditional (wrt the frailty). This can be briefly mentioned so that repetition is not needed.

Throughout the main text and supplement, we have updated the HR abbreviation to indicate whether the unconditional, crude HR is being reported (cHR) or the conditional, adjusted HR is being reported (aHR). Models that produced an aHR always included a random intercept at the individual level.